# Understanding Worker Well-Being Relative to High-Workload and Recovery Activities across a Whole Day: Pilot Testing an Ecological Momentary Assessment Technique

**DOI:** 10.3390/ijerph181910354

**Published:** 2021-10-01

**Authors:** Raymond Hernandez, Elizabeth A. Pyatak, Cheryl L. P. Vigen, Haomiao Jin, Stefan Schneider, Donna Spruijt-Metz, Shawn C. Roll

**Affiliations:** 1Chan Division of Occupational Science and Occupational Therapy, University of Southern California, Los Angeles, CA 90089, USA; beth.pyatak@usc.edu (E.A.P.); vigen@usc.edu (C.L.P.V.); sroll@usc.edu (S.C.R.); 2Dornsife Center for Economic & Social Research, University of Southern California, Los Angeles, CA 90089, USA; haomiaoj@usc.edu (H.J.); schneids@usc.edu (S.S.); dmetz@usc.edu (D.S.-M.); 3Keck School of Medicine, University of Southern California, Los Angeles, CA 90033, USA; 4Department of Psychology, University of Southern California, Los Angeles, CA 90089, USA

**Keywords:** workload, recovery, ecological momentary assessment, type 1 diabetes, workweek, healthy work design and well-being, future of work

## Abstract

Occupational health and safety is experiencing a paradigm shift from focusing only on health at the workplace toward a holistic approach and worker well-being framework that considers both work and non-work factors. Aligned with this shift, the purpose of this pilot study was to examine how, within a person, frequencies of high-workload and recovery activities from both work and non-work periods were associated with same day well-being measures. We analyzed data on 45 workers with type 1 diabetes from whom we collected activity data 5–6 times daily over 14 days. More frequent engagement in high-workload activities was associated with lower well-being on multiple measures including higher stress. Conversely, greater recovery activity frequency was mostly associated with higher well-being indicated by lower stress and higher positive affect. Overall, our results provide preliminary validity evidence for measures of high-workload and recovery activity exposure covering both work and non-work periods that can inform and support evaluations of worker well-being.

## 1. Introduction

The future of work is projected to bring with it several novel and complex concerns, prompting the National Institute for Occupational Safety and Health (NIOSH) to develop and promote holistic and transdisciplinary approaches to worker health and well-being [1]. Advances in technology are expected to continue influencing how workers engage in work and their workplace, including an increased frequency of remote work [2] and a greater degree of job automation [3]. The COVID-19 pandemic has drastically increased the pace at which remote working arrangements have been adopted, with roughly 35% of employees working from home at the height of the pandemic [4]. Anticipated consequences of these changes include blurring of work–life boundaries, greater expectation of work availability outside of standard hours, and increased potential for isolation [3,5]. To address these consequences, NIOSH implemented the Total Worker Health^®^ (TWH) program, which aims to protect workers from work-related safety and health hazards while also promoting their health more generally [6]. NIOSH also proposed a new worker well-being framework that reflects this holistic paradigm shift and supports consideration of non-work factors that could impact worker health [7]. This expanded approach to worker well-being integrates work and non-work health promotion, acknowledging that both personal and occupational risk factors may contribute in an additive fashion to health and safety outcomes [8].

Workload and recovery are frequent areas of worker health research that could benefit from this more holistic and integrated perspective on supporting worker health and well-being. Workload is most commonly associated with working adults, who are exposed to higher amounts of workload from their paid employment [9,10] than are other populations, such as retirees [11]. However, workload can be more broadly considered as the cost (e.g., fatigue, stress, illness) of performing any tasks [12], whereby high-workload activities can be experienced in both work and non-work times. For instance, caregiving is a potential source of non-work, high workload [13]. Recovery is the process that repairs the negative strain effects stemming from exposure to stressors [14], a concept that is often applied to leisure time or sleep but can also be found in activities throughout a workday. Although workload and recovery apply to both work and non-work activities that individuals engage in throughout a day, prior studies have typically partitioned the investigation of exposures by examining high workload at work [15] and recovery during non-work time [10]. As such, excessive exposure to high-workload tasks from work has frequently been associated with poorer psychological and physical well-being [15,16,17], while greater engagement in recovery after work has been associated with positive health and well-being outcomes [18].

Despite conceptual support using a siloed approach, the TWH philosophy and emerging holistic frameworks indicate that worker well-being may best be understood by examining activity engagement across both contexts to more fully capture the combined effects of workload and recovery [8]. That is, just as breaks taken during work are likely important to consider as part of total recovery, high-workload engagements during non-work time are also likely essential for measuring total workload.

We are, in essence, suggesting the need for an expanded scope within the Effort-Recovery model that accounts for both work and non-work activity engagements. According to this model, which was in part based on exercise literature, exerting effort (i.e., exposure to workload) during work leads to load reactions, acute changes to psychobiological systems that are adaptive in the short term in that they help in meeting task demands [9]. Adequate time for recovery, defined as return of psychobiological systems to baseline after exertion ceases, allows for reversal of the load reactions and a return of psychobiological systems to pre-demand levels [9]. With sufficient recovery, fatigue and other manifestations of load reactions are reduced [19]. Insufficient recovery however can lead to impaired well-being [19]. Much work has been investigating the nuances of the relationships between effort/recovery and well-being. For instance, job autonomy, role clarity, and social support have been theorized as potential moderators of the link between workload and well-being [20]. Here, however, we focus on the assertion that while the Effort-Recovery model traditionally focuses on work contexts, consideration of both work and non-work activity engagements may be needed to capture full exposure to effort (i.e., workload) and recovery. This would create a fuller picture of how well-being emerges compared to a siloed approach, where part of the picture is missing.

The purpose of this pilot study was to examine the feasibility of ecological momentary assessment (EMA), i.e., repeated sampling of behaviors and experiences in real time and in natural environments [21], as a method for investigating worker health and well-being as part of this expanded scope. Specifically, we collected EMA data five to six times a day for 14 days to examine how the frequencies of workers’ exposure to high-workload and recovery activities, across both work and non-work periods over whole days, were associated with same day well-being measures. As shown in Table 1, we expected that valid EMA data would result in the following: (1) frequency of high-workload activities across a whole day would be positively associated with end of day workload [9] and negatively associated with well-being [15], and (2) frequency of recovery activities across a whole day would be negatively associated with workload [9] and positively associated with well-being [22,23].

## 2. Materials and Methods

### 2.1. Study Overview

We analyzed data collected as part of a multisite study investigating the relationship between function, emotion, and blood glucose in adults with type 1 diabetes (T1D), described in greater detail in the published study methodology paper [24]. Participants aged 18–75 with T1D were recruited from three clinical sites, two in Los Angeles and one in New York, through mailings, phone calls, email invitations, and health provider referrals. Due to similarities in the implications of having chronic conditions, we anticipated that our T1D population may be somewhat similar to other worker populations with chronic conditions, thereby increasing the potential generalizability of our results (discussed further in Section 4.3). Only participants that identified as workers were included in analyses. Participants completed a baseline survey battery, 14 days of EMA data collection with 5–6 surveys per day, and a follow-up survey battery. The data collection procedures were approved by the University of Southern California Institutional Review Board. Subjects provided informed consent to participate in the study electronically through the REDCap e-consent framework [25].

### 2.2. Measures

For approximately two weeks, EMA surveys were administered on smartphones with up to 6 momentary surveys per day at 3-h intervals, using the Mobile EMA application (mEMA: ilumivu.com). Relative to other data collection methods such as end of day surveys or time diaries, an advantage of EMA is reduced recall bias since data are reported immediately after experiences of interest have occurred [21]. At each time point, participants identified the type of activity they were presently engaging in and rated state-level indicators of well-being, and, at the last assessment of each day a subjective assessment of overall workload was obtained (Table 2). Note that measures listed in Table 2 are only a subset used for analyses in this paper. For a full list of the items administered, please refer to the methods paper for the overarching study [24].

#### 2.2.1. Activity Exposure

At the beginning of every EMA survey, participants indicated the activity in which they were engaged immediately before the survey (sampling approach) [21] as opposed to reporting an activity that was engaged in for the greatest length of time in the hours prior (coverage) [21]. They were forced to choose only one activity option and prompted to choose the main or most important activity if several of them were being engaged in simultaneously. The sampling approach was used rather than a coverage method to minimize subjective selection of activities by the participants and further reduce recall bias. The activity question was adapted from a prior EMA study [26]. Possible activity responses were based on a taxonomy of activities created by occupational therapists based on their practice framework [27] and decided upon with expert review by occupational scientists. During baseline training, participants were given examples of types of activities within each category; these detailed examples were not presented with the answer choices to reduce word clutter on the phone screen.

The responses were conceptually sorted into the two primary constructs of high-workload and recovery based on inferences made from prior literature, the former including both work and non-work strenuous activities and the latter including leisure and rest activities. To do so, we emulated an approach used in prior studies where activities are sorted into categories based on their typical characteristics (e.g., watching television typically represents a “sedentary” behavior) [28].

High-workload tasks included “work/school” and “caring for others”. “Work/school” was placed in this category because of the large body of literature conceptualizing work as a demanding task that requires recovery [9,10]. Caregiving is one of the few sources of non-workplace demands that has been frequently investigated [13]. While many studies report adverse effects of caregiving [29,30], others have highlighted the benefits [31,32]. One broadly supported idea is that caregiving may have positive effects as long as workload is not too excessive (e.g., time demands being too high) [29,33]. We argued that given how often caregiving is conceptualized as a demand source that requires recovery periods [34], it conceptually belongs to the high-workload category. This classification was not meant to ignore the potential positive aspects of caregiving such as companionship and a sense of fulfillment [31], but to acknowledge the demands that often come with providing care for other individuals.

Recovery activities included “fun/play/leisure”, “socializing”, “relaxing/chilling”, and “sleeping/napping”. “Relaxing/chilling”, and “sleeping/napping” were counted as recovery because both are often accompanied by increases in parasympathetic activity, which has been conceptualized as a defining feature of rest [35]. “Fun/play/leisure” and “socializing” were also subsumed under recovery because both are forms of leisure [36], which is an often-cited form of recovery [10].

Any activity choice that did not fit into high-workload or recovery groupings was placed into an “unclassified” category. “Doing housework/errands” was one activity placed into this category because there is limited published research to provide a clear interpretation of these tasks. For example, “cleaning” is considered restful by some and demanding by others, making it difficult to define as either high-workload or recovery without further understanding an individual’s context. Participants also had an option to specify “other activities” not covered by the a priori activity response options using an open-ended response. These responses were assigned to one of the three categories during a post hoc review process. For example, an open-ended response “vigorous exercise” would fit under “high-workload” because of the short-term demand, though it may have a long-term benefit.

#### 2.2.2. Perceived Workload

We administered additional measures explicitly tapping participants’ perceptions of workload at the end of each day to validate that our EMA derived high-workload and recovery activity frequency measures had the expected associations with end-of-day workload measures. Survey questions were selected from validated global measures or those that had been used successfully in previous EMA studies. Task load is the level of difficulty an individual encounters when executing tasks [37] and was assessed with an adapted version of the National Aeronautics and Space Administration Task Load Index (NASA-TLX) [38]. Like the original TLX, we also used six items addressing different contributors to task load, but changed the period being inquired about to a whole day rather than a particular task. In the nursing literature, there has been some evidence supporting the validity of use of the TLX items to cover entire periods instead of particular tasks, specifically whole work shifts [39]. The NASA-TLX provides an overall task load score based on the summation (or equivalently, averaging) of an individual’s perception of mental demand, physical demand, temporal demand, performance, effort, and frustration level [12]. While a weighting scheme accounting for individual variation in the extent to which each dimension of demands contributes to overall task load was utilized for the original TLX, a review found that raw sums (or equivalently, simple average) appeared just as valid [12]. Total hours worked served as a behavioral measure of general workload over the day.

#### 2.2.3. Well-Being Experiences

Participants’ momentary well-being experiences were measured using a variety of state-level factors including affect, working memory, stress, fatigue, and pain—all of which are core well-being concepts encapsulated in the health status section of NIOSH’s new Worker Well-being Questionnaire (NIOSH WellBQ) [40]. For positive and negative affect, items were taken from the “Stress and Working Memory” (SAWM) study [41], which were adapted from the Positive and Negative Affect Schedule (PANAS) [42] by changing the period referenced (e.g., asking about emotions in the moment) and the mood items used. The items for fatigue and pain were taken from prior a EMA study [43], where they were derived from the Brief Fatigue Inventory [44] and Brief Pain Inventory, respectively [45]. Our item for stress was used in a prior EMA study [46].

#### 2.2.4. Scoring of Measures

Various scoring approaches were used for different measures. For single item assessments of stress, fatigue, pain, and work hours, there was no score calculation step. The workload, affect, and activity exposure measures however did involve the calculation steps described prior. Scoring algorithms for workload, positive/negative affect, and high workload activity frequency were decided prior to reviewing the dataset. For calculation of recovery frequency however, prior to looking at the data, a scoring approach was used where “relaxing” and “leisure” were treated as separate forms of recovery. After review of the initial draft, one of the co-authors commented that collapsing relaxing and leisure under the umbrella of recovery made more sense. Thus, scoring for recovery exposure was changed to include both relaxing and leisure activities.

### 2.3. Statistical Analyses

To calculate daily activity frequencies within the high-workload and recovery categories, we divided the number of times people reported engaging in a particular activity type by the number of EMA assessments taken [47]. For example, if a participant completed six surveys in a day and reported engaging in activities within the high-workload category during three surveys, then the relative frequency of high-workload for that day would be 3/6 = 0.5. Frequencies were only calculated for days where participants completed four or more surveys.

We conducted intra-person correlational tests to investigate the associations of our EMA derived daily activity frequencies with end of day workload and well-being measures. In addition to daily activities, we hypothesized that weekends would have lower high-workload and higher recovery activity frequencies than did days during a traditional 5-day work week. Mixed-effects modeling was used with day of the week as the only predictor, the intercept as the dependent variable’s random effect, and frequency of activity groupings (e.g., high-workload and recovery) as the dependent variable. Models were run with the “lme4” package in the statistical software R [48]. Mixed modeling was used to account for the nested nature of the data (multiple days nested in an individual) while testing hypotheses involving multilevel categorical predictors (e.g., day of week) [49]. We chose Sunday as the reference for ease of comparison, as we anticipated that it would have the greatest frequency of recovery and lowest frequency of high-workload activities. Given that *p* values of parameters from mixed models may be biased because of the inherent uncertainty in the degrees of freedom to specify for calculations [50], especially when the data structure is complex or the dataset is unbalanced, we used bootstrapping to derive standard errors for the parameters and perform significance testing [50,51]. Bootstrapping was also used to account for potential non-normality in the mixed model residuals using the “method = boot” argument available in the “lme4” package. The decision to perform mixed models was done a priori, to allow for comparison of the mean values of high workload and recovery across different days of the week, and have a corresponding indication of statistical significance. The initial plan was to just present mean values of high workload/recovery/unclassified by day of week, but it seemed unsatisfactory to not be able to comment on statistical significance of the differences.

The “power.rmcorr” function in R was used to estimate the statistical power to detect repeated measures correlations [52]. Based on prior daily diary research, we assumed an intraclass coefficient of 0.5 for the nesting of repeated observations in individuals [53,54] for the power calculations. A priori sample size calculations suggested that a sample size of 30 with 13 observations per individual would yield 82% power to detect within-person effect sizes of 0.15 with α = 0.05 (a small effect as per Cohen’s conventions [55].

## 3. Results

Analyses were conducted on data from a total of 45 participants from the multisite study who identified as workers. The participants primarily worked full-time, had an average age of 40.1 (SD = 12.7) years, and were heterogenous with respect to gender, ethnicity, education, and annual income (Table 3). Participants were not required to report their vocations, but occupations reported include lawyer, engineer, housekeeper, teacher, and security guard. The median EMA completion percentage was 92%, and four or more EMA surveys were completed on 83% of all data collection days across the participants. The average number of days meeting the minimum threshold for inclusion of four EMA surveys was 12.5 days per participant. The final dataset included a total of 3352 EMA datapoints across 564 valid days. In post hoc power analyses, our sample of 45 participants with 12.5 observations per individual yielded 84% power to detect within-person effect sizes of 0.13 with α = 0.05.

The distribution of responses within each of the activity types is provided in Table 4. Approximately half of the datapoints were equally split between work/school (25.8%) and relaxing/chilling (22.0%). One-third of the responses were split between sleeping/napping (13.0%), housework/errands (11.8%), and self-care activities (11.2%). A total of 920 responses were categorized as high-workload, 1452 as recovery, and 980 as unclassified. High-workload activities had a lower average daily relative frequency (0.28, SD = 0.25) compared to the average daily relative frequency of recovery activities (0.43, SD = 0.25).

Full between and in-person correlation matrices are shown in Appendix A and Appendix B respectively. Note that the correlations shown in Table 5 are only the subset of associations from Appendix A that were of interest in this paper, along with their confidence intervals. We did not address between-person correlations in this paper, as the within-person context was of interest, and we only had sufficient power to investigate within-person questions and not between-person ones. Thus, between-person associations shown in Appendix A should be cautiously interpreted, and were included to highlight the distinction among between- and within-person relationships.

Except for pain, all associations among the daily relative frequencies for high-workload and recovery activities with measures of workload and well-being were in line with our hypotheses (Table 5). High-workload activity frequency had a moderate positive association with overall task load (r = 0.42), work hours (r = 0.31), and stress (r = 0.34), whereas recovery activity frequency had a moderate negative association with overall task load (r = −0.37) and stress (r = −0.34). Fatigue, positive affect, and negative affect were weakly associated with the two categories, and the two groupings had opposite directions for each. Unclassified activity frequency had only a weak negative association with work hours (r = −0.17), with no other associations with workload or well-being.

The daily relative frequencies of high-workload and recovery activities had nearly opposite patterns when examining their distributions across days of the week (Figure 1). Recovery activities accounted for a clear majority of the activities (i.e., >0.50) on the weekend and were slightly more prevalent than high-workload activities on Mondays and Fridays. High-workload and recovery activity frequencies were balanced through the middle of the week. Mixed model results for the daily activity frequency measures were consistent with our hypotheses. High-workload activities were more frequent on weekdays compared to Sunday, and recovery activities were less frequent on weekdays compared to Sunday (Table 6). The mean frequency of unclassified activities was only different from Sunday on Monday through Wednesday.

Measures of well-being showed less consistent patterns of mean level differences across the days of the week (Figure 2). Stress was the only state-level well-being measure that demonstrated fluctuations across days of the week, with a pattern comparable to that of high-workload activity frequency. A slight upward trend in fatigue was noted across the week, and participants generally reported low levels of pain, minimal negative affect, and high positive affect, measures that all remained relatively stable across the week. 

## 4. Discussion

### 4.1. Principal Findings

Our results supported the usefulness of our EMA-derived daily activity frequency measures, with one piece of supporting evidence being our observation of weekend versus weekday differences in theoretically expected directions. Though not hypothesized a priori, mean high-workload activity frequency steadily increased from Monday to Wednesday and then decreased on Thursday and Friday. Conversely, mean recovery activity frequency steadily decreased from Monday to Wednesday and increased on Thursday. This is consistent with the finding that fatigue, a possible proxy measure of the amount of workload relative to recovery experienced, was found to peak midweek and then decrease for workers [56]. Whole day workload and recovery by day of week has not yet been formally examined.

Results of within-person correlation tests also generally provided evidence supporting the trustworthiness of our daily activity measures. Frequency of unclassified activities had a significant but weak association with work hours (r = −0.17, *p* = 0.001). This relationship may have been driven by slight increases in engagement in unclassified activities (e.g., housework/errands) on the weekends when high-workload activities decreased. Despite this association, unclassified activities were not found to be related to measures of task load or any of the well-being factors, further indicating that deeper examination of contextual factors underlying these activities is needed to better understand their impact either on demand or recovery as a component of a worker’s day.

In contrast to the lack of association in unclassified activities, daily frequencies of high-workload and recovery activities followed the hypothesized associations with well-being measures. This finding provides preliminary support that an EMA derived whole day metric of activities may be useful for exploring the holistic impact of both work and non-work periods. Across the correlations among high-workload and recovery activities with measures of workload and well-being, 86% were consistent with our hypotheses providing strong evidence of convergent validity. Convergent validity is considered to be adequate if at least 75% of hypothesized correlations are empirically supported [57,58].

Given the number of statistical tests conducted, there is a chance of spurious associations. We conducted approximately 39 statistical tests, which included21 within-person correlation tests and 18 beta parameter tests in mixed models. This total includes tests for unclassified activities that were done for exploratory purposes as opposed to hypothesis testing. Assuming an alpha of 0.05, approximately 2 (39 × 0.05) of the “significant” findings as per confidence intervals may have been spurious associations. Given how small this number is however relative to the number of associations we found aligned with our hypotheses, the overall findings of this pilot work provide strong preliminary evidence in supporting the evaluation of within-person relationships among workload/recovery across both work and non-work settings and measures of worker well-being.

### 4.2. Implications for EMA in Worker Health Research

While the relationship between workload/recovery and well-being has long been researched [15,18], the novelty of this study lies in the development of methods to assess high-workload and recovery activity exposure across both work and non-work settings using EMA methods. Studying the experience of high-workload and recovery activities by considering both work and non-work contexts matches well with NIOSH’s intent to support workers of the future through its TWH initiative [1,6,7,8]. When workload and recovery are examined only in work or non-work scenarios, researchers are likely to attain an incomplete picture of total exposure and a fragmented depiction of how well-being outcomes arise. The actual relationships found between whole day activity frequencies and various well-being measures were generally in the expected directions. This provides preliminary evidence that measures with a coverage period comprising both work and non-work can provide meaningful information.

From a psychometric standpoint, no prior studies have formally validated daily activity frequencies calculated from a single EMA activity engagement item. Our results imply that our measurement approach, involving classification of a broad range of activity choices and tallying of the number of reports of particular activities in a day, may be appropriate for specific activity types. Activity frequencies derived from a single EMA activity engagement item would likely be particularly useful in studies already planning to administer multiple EMA signals daily for several days. With minimal additional burden to participants, adding a single activity engagement item would allow researchers to gauge time use in an activity believed to be relevant to the primary construct of interest. For instance, in EMA studies investigating links between momentary patient reported outcome data and ambulatory physiological measures (e.g., blood pressure, blood glucose) in workers, a single activity item could potentially allow for exploration of questions such as how time spent in particular activities (e.g., work or leisure) is associated with the values of physiological measures on the same day.

EMA based time use measurement can also potentially be used by organizational leadership to approximately gauge time employees spend in high-workload and recovery activities over entire days, especially if they are already using EMA to measure other variables of interest such as ergonomic positioning or mood [59]. For instance, employers can work to identify the circumstances that contribute to excessive exposure to high workload relative to recovery. This imbalance may occur across all workers during certain days of the week or seasons of the year. Employers can also investigate relationships between whole day high workload/recovery and other within person measures of interest such as productivity or frequency of job errors on a particular day.

The utility of measuring daily workload directly (e.g., with the TLX) compared to assessing engagement in high-workload activities through EMA requires further research. Theoretically, the former requires that participants make a judgement of how demanding tasks across the day felt, while the latter requires them to just report activities that they engaged in, from which high workload exposure is calculated. Which is more useful may depend on the research question (e.g., if judgement-based vs. activity- report derived workload ratings are of interest), and which is found to have a greater relationship with the outcome(s) of interest (e.g., stress).

### 4.3. Limitations and Future Directions

Findings of this study should be considered within the context of this work as a pilot study. This includes an acknowledgement that this work is not intended to indicate conclusive relationships among whole-day exposures and well-being, but to serve as preliminary support for a process that could be used to examine these relationships.

Moreover, it should be noted that data collection was limited to individuals who had type 1 diabetes and were experiencing the COVID-19 pandemic and its associated social distancing requirements during data collection. Replication of this process with larger samples and within other worker populations and settings is needed to confirm our preliminary findings as more conclusive evidence of the relationships suggested in this study. Due to similarities in the implications of having chronic conditions, we anticipate that our T1D population may be somewhat similar to other worker populations with chronic conditions, suggesting the potential for future studies to generalize our study findings. For instance, the Chronic Disease Self-Management program, a widely used intervention for chronic conditions, assumes that patients with different chronic diseases have similar self-management problems and disease related tasks [60], which can include following complex treatment regimens, self-monitoring one’s condition(s), and making decisions about when/how to receive professional treatment [61]. All these responsibilities are additional potential sources of workload, which may in part explain why management of chronic conditions in general is frequently perceived as intrusive to people’s daily lives [62]. As additional evidence of the commonalities between chronic conditions, research is often conducted on populations with any chronic condition [63,64]. Nearly 45% of all Americans suffer from at least one chronic condition [65]. While our T1D population is potentially similar to populations with chronic conditions generally, they may have more commonalities with people with type 2 diabetes (T2D). Thus, our findings may generalize more easily to people with T2D as compared to populations with chronic conditions generally. T1D and T2D have different etiologies, and T2D is the much more common of the two, with a prevalence of 8.5% in the U.S as of 2017 as compared to 0.5% for T1D [66]. However, they share similar management approaches (e.g., insulin therapy [67]) which can often serve as sources of distress [68]. In research studies, participants with both T1D and T2D are often pooled together [67,68].

In addition to limitations inherent in a pilot study, because activity engagement was a secondary measure within the primary study, precision for the EMA activity item was perhaps limited. Specifically, to minimize participant burden only nine broad activity choices were provided, and no follow-up questions were asked. The possible realm of activities are likely much more nuanced and capturing contextual factors and relative meaning to each individual might increase the ability to more precisely categorize activities as high-workload or recovery. One approach that may be worth testing in future studies is asking participants to first choose a general category of activities that best matches what they had been doing immediately prior to the survey (e.g., housework), and then use branching logic to ask about the specific activity done (e.g., washing dishes or vacuuming). This would allow for more detailed participant activity reports while reducing screen clutter. An additional semantic differential question allowing participants to rate the activity on a spectrum from high-workload to recovery could begin to fill the gap in understanding individual perspectives on activities, particularly those falling within our unclassified category.

Because the precision for our EMA activity item was limited we could not account for individual variation in context, and misclassification during conceptual sorting of activity types into high workload or recovery may have occured. For instance, if a participant reported “fun/play/leisure” this activity would always fall under the recovery category. However, “fun/play/leisure” in the form of a friendly board game could have been high-stress if it had evolved into cutthroat competition. As such, implicit in our conceptual sorting scheme was a certain degree of error that likely created a deviation from true high-workload/recovery activity exposure. A benefit of this rough sorting heuristicwas that it only required a single EMA item asking about type of activity engagement, and our results indicate that the sorting was good enough to capture relationships with the well-being measures. In future studies, increased accuracy could be achived using a more refined sorting of activities into high-workload and recovery bins may be possible if, in addition to a question about type of activity engaged in, there is also a rating of the workload associted with that activity, as suggested in the prior paragraph. The downside of this approach is the added respondent burden.

EMA measures are not formally validated as often as one time cross-sectional measures [69], so another study limitation to consider is that evidence of validity for EMA assessments used in this paper comes in the less definite forms of being adapted from validated global measures, and/or being successfully used in prior studies. While the approach of arguing for the validity of EMA assessments based on derivation from an establish global measure is often taken [43,70], it is also common for EMA studies to rely on face validity for some items [46,71]. The full versions of global measures often cannot be used in the EMA context due to excessive length, so adaption to shorter versions is often required [70]. While some work has been done to formally validate these shortened versions [72], such efforts often focus on validation in the between-person context and not within-person [69], like in our work.

Worker well-being is a broad multi-dimensional concept, and this pilot study did not address all aspects of well-being. We were limited in our choice of well-being measures to items included as part of the overarching study investigating the relationship between a variety of variables and blood glucose. Well-being variables that were included were general enough to be appliable to a large segment of workers, as evidenced by their inclusion in the NIOSH WellBQ, but do not comprehensively capture well-being. Future studies may be needed to investigate how high-workload and recovery activity exposure over whole days relate to other aspects of well-being (e.g., sleep quality, life satisfaction) not covered in this study.

Two further limitations are noted in our statistical approach. Firstly, within the context of worker well-being, researchers and practitioners are commonly interested in examining between-person questions, such as how workers with different job classifications and/or work schedule arrangements differ in their mean exposure to high workload and recovery activities. This pilot study was designed to provide preliminary support for measuring within-person relationships among activities and well-being, and did not examine between-person contexts in part because we were insufficiently powered to do so. Given that our findings supported general within-person hypotheses, future studies could examine feasibility for using EMA to create between person-level outcomes (e.g., averages over the entire study period) that could support between-person analyses. For instance, future studies might be able to conduct concurrent validity testing of study period averages of the daily activitiy frequencies (between-person metrics) by comparing their values to those attained from full questionnaires, such as the newly released NIOSH WellBQ. Secondly, we used the standard approach in longitudinal data analyses of assuming that our missing data were missing at random [73], which may have biased our results. There is a possibility that data was not missing at random. For instance, high workload engagements may have been undercounted because some participants may be less likely to answer surveys during those activities. Work on modeling data not missing at random in the longitudinal context is ongoing [73].

A final limitations in this pilot work is that our results did not allow us to assess our theoretical assertion that, when workload and recovery are examined only in work or non-work scenarios, researchers are likely to attain an incomplete picture of total exposure and a fragmented depiction of how well-being outcomes arise. Instead, our results provide evidence supporting one potential way to measure workload/recovery across both work and non-work. Future research, posssibly using the EMA assessment used here, is needed to compare how/if whole day and work specific measures of workload/recovery differ in their associations with well-being. Our theoretical assertion would be supported if the whole day version has stronger associations with well-being relative to assessment only of work periods.

## 5. Conclusions

In this pilot study, we found that measures of the frequency of high-workload and recovery activities over a whole day generally had the expected associations with same day well-being measures. More frequent engagement in high-workload activities was associated with worse well-being experiences including higher stress and lower positive affect. Conversely, greater recovery activity frequency was mostly associated with better well-being including lower stress and higher positive affect. Activity frequency measures also had anticipated associations with day of the week whereby high-workload activities peaked during the middle of the week and recovery activity frequency was more common on the weekends. Our results provide preliminary validity evidence for measures of high-workload and recovery activity exposure across a whole day that covers both work and non-work periods. This assessment of both work and non-work factors may facilitate strategies for identifying job classifications, work schedules, or individual workers who may be most at-risk for negative impacts on well-being. Application of these EMA techniques may serve as a basis for employers to develop policies or programs to support health and well-being across their workforce.

## Figures and Tables

**Figure 1 ijerph-18-10354-f001:**
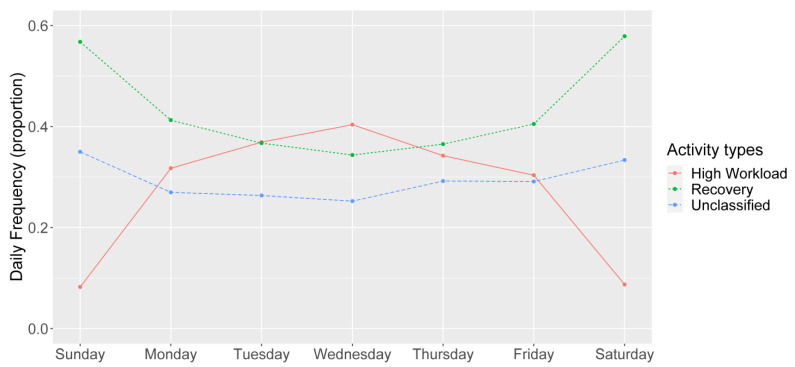
Distribution of activity frequencies, by day of week.

**Figure 2 ijerph-18-10354-f002:**
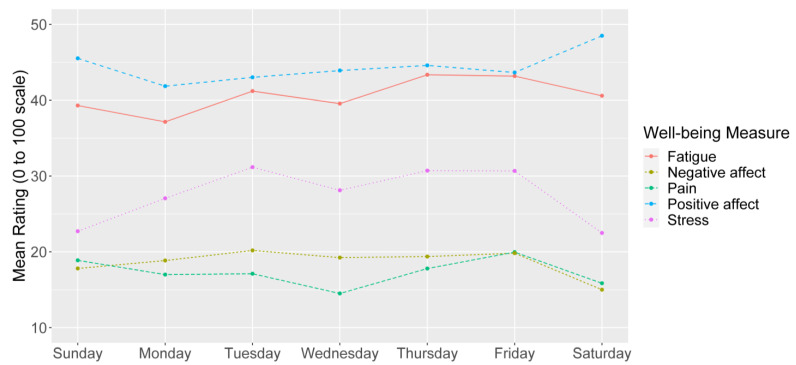
Average ratings of state-level well-being, by day of week.

**Table 1 ijerph-18-10354-t001:** Hypothesized intra-person correlations between relative activity frequency and measures of workload and well-being.

	Workload			Well-Being			
	Overall Task Load	Work Hours	Stress	Fatigue	Pain	Positive Affect	Negative Affect
High workload activity frequency	+	+	+	+	+	−	+
Recovery activity frequency	−	−	−	−	−	+	−

−: negative correlation; +: positive correlation.

**Table 2 ijerph-18-10354-t002:** Ecological momentary assessment measures administered over up to 14 days of data collection.

Construct	Item(s)	Response Option(s) ^1^	Time
Activity			
Activity type	What were you doing right before starting this survey?	Work/school activities (e.g., paid labor, volunteer work, and studying) Traveling (e.g., driving, riding in a car, walking) Relaxing/chilling (e.g., passive leisure like watching Netflix, listening to music) Sleeping/napping Socializing (e.g., talking with friends/family) Caring for myself (e.g., eating, dressing, bathing, toileting, personal grooming) Caring for others (e.g., caring for your children and pets, if you’re caring for others as part of work this counts as “work”) Doing housework/errands (e.g., paying bills, washing dishes and clothes, exercising for health) Fun/play/leisure activities (e.g., active leisure like exercising for fun, video games, reading for fun) Other (If chosen, please specify)	All survey times
Well-Being			
Stress	How stressed are you right now?	0 (Not at all stressed) to 100 (Extremely stressed)	All survey times
Fatigue	At this moment, how tired do you feel?	0 (Not at all) to 100 (Extremely)	All survey times
Pain	At this moment, how much bodily pain do you have?	0 (None) to 100 (Extreme pain)	All survey times
Positive affect	4 items: Average of mood ratings for “happy”, “content”, “enthusiastic”, “excited”	For each mood, 0 (not at all) to 100 (extremely)	All survey times
Negative affect	4 items: Average of mood ratings for “tense”, “upset”, “sad”, “disappointed”	For each mood, 0 (not at all) to 100 (extremely)	All survey times
Workload			
Task Load	Average of 6 NASA-TLX items asking about mental demand, physical demand, time pressure, effort, performance satisfaction, and frustration for activities over the whole day.	0 to 100 sliding scale for each item, and the overall task load score	End of day
Work hours	(If worked) About how many hours did you work?	Hours, whole number, 0 to 24	End of day

^1^ Activity examples were not in the actual item, but were explained during training and listed in a manual provided to participants.

**Table 3 ijerph-18-10354-t003:** Demographic characteristics of our working sample (*n* = 45).

Characteristic	*n*	Mean (SD) or Percent (%)
Age (years)	45	40.1(12.7)
Gender		
Male	21	47%
Female	24	53%
Ethnicity		
White	20	44%
Latino/x	10	22%
African American	7	16%
Multi-ethnic	4	9%
Other	4	9%
Employment status		
Full-time	35	78%
Part-time	10	22%
Education		
High school grad or less	5	11%
Some college, no degree	6	13%
Associate’s degree	1	2%
Bachelor’s degree	18	40%
Graduate degree	15	33%
Annual household income		
<$50,000	9	20%
$50,000–$99,999	10	23%
≥$100,000	15	34%
Do not wish to provide	8	18%
Do not know	3	7%

**Table 4 ijerph-18-10354-t004:** Frequency distribution of activity types across all datapoints (*n* = 3352).

Activity Type	Frequency (%)
Work/school activities	866 (25.8%)
Relaxing/chilling	739 (22.0%)
Sleeping/napping	437 (13.0%)
Doing housework/errands	396 (11.8%)
Caring for myself	374 (11.2%)
Fun/play/leisure activities	191 (5.7%)
Traveling	160 (4.8%)
Socializing	85 (2.5%)
Caring for others	54 (1.6%)
Other	50 (1.5%)

**Table 5 ijerph-18-10354-t005:** Within-person correlations among daily frequency of activity categories and measures of workload and well-being. Values in parentheses represent 95% confidence intervals.

	Overall Task Load	Work Hours	Stress	Fatigue	Pain	Positive Affect	Negative Affect
High workload activity frequency	0.42, (0.34, 0.49)	0.31, (0.21, 0.4)	0.34, (0.26, 0.41)	0.11, (0.03, 0.2)	0.02, (−0.07, 0.11)	−0.15, (−0.23, −0.07)	0.21, (0.13, 0.29)
Recovery activity frequency	−0.37, (−0.44, −0.29)	−0.14, (−0.25, −0.04)	−0.34, (−0.41, −0.26)	−0.10, (−0.19, −0.01)	−0.06, (−0.14, 0.03)	0.14, (0.06, 0.23)	−0.22, (−0.3, −0.13)
Unclassified	−0.08, (−0.16, 0.01)	−0.17, (−0.27, −0.07)	−0.02, (−0.11, 0.07)	−0.02, (−0.11, 0.06)	0.04, (−0.04, 0.13)	0.02, (−0.07, 0.1)	−0.01, (−0.1, 0.08)

**Table 6 ijerph-18-10354-t006:** Day-of-week differences in activity frequencies. Values in parentheses represent 95% confidence intervals.

Day	Number of Observations	High-Workload Frequency Beta, (95% CI)	Recovery Frequency Beta, (95% CI)	Unclassified Frequency Beta, (95% CI)
Sunday	89	Reference	Reference	Reference
Monday	87	0.24, (0.18, 0.29)	−0.16, (−0.22, −0.1)	−0.08, (−0.14, −0.02)
Tuesday	106	0.29, (0.23, 0.34)	−0.21, (−0.28, −0.15)	−0.07, (−0.13, −0.02)
Wednesday	105	0.32, (0.26, 0.38)	−0.24, (−0.3, −0.18)	−0.09, (−0.14, −0.03)
Thursday	107	0.26, (0.19, 0.31)	−0.2, (−0.26, −0.14)	−0.05, (−0.11, 0.01)
Friday	104	0.22, (0.16, 0.28)	−0.17, (−0.22, −0.11)	−0.05, (−0.11, 0.00)
Saturday	78	0.00 (−0.06, 0.06)	0.01, (−0.05, 0.08)	−0.01, (−0.07, 0.05)

## Data Availability

The data that support the findings of this study are available from the corresponding author, R.H., upon reasonable request.

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
