# Peer review of "Understanding Worker Well-Being Relative to High-Workload and Recovery Activities across a Whole Day: Pilot Testing an Ecological Momentary Assessment Technique"

_ijerph, 2021, doi:10.3390/ijerph181910354_

Round 1

Reviewer 1 Report

Thank you for the opportunity to review this manuscript. This is an area of current interest. Please see my comments for your consideration:

MATERIALS & METHODS:

2.1 Study overview: Are you able to report the number of sites (workplaces) involved in this research? Which sector/industry do they represent? How many workers were recruited? What was the recruitment strategy - how were the recruited sites and participants selected? Was there attrition - response rate? 

2.2 Measures: It would be useful for the reader to report the number of items for each of the measures. How many items from NASA-TLX, NIOSH questionnaire, PANAS, SAWM and  prior EMA studies? How many items were there in total? How long did it take on average to complete a survey? Would it be possible to include your survey instrument in this manuscript as a supplement? 

DISCUSSION: There is good discussion about the utility of the EMA techniques discussed in this research However, it would be good to know the information requested above to comment on its feasibility. 

Reviewer 2 Report

This is a very original and methodologically interesting paper. It is well-written, concise, and would be a contribution to the TWH literature because of the methodologic approach. I have a number of major questions and recommendations for improving the paper, presented below.

The authors set out to address how workload and recovery contribute to well-being of workers and propose that activity engagement across both work and non-work contexts will more fully capture the combined effects of workload and recovery. They state that the “purpose of this study was to examine how the frequencies of high-workload and recovery activities over a whole day were associated with same day well-being measures.” 

This is a pilot study with findings that are “preliminary”, as acknowledged by the authors. It should be labeled as a pilot study in the title and throughout the paper, mainly because I think the interpretation of their data should be more cautious given the size of the study and the number of comparisons being made without clear statements and testing of hypotheses. It feels more exploratory than conclusive. Efforts should be made to not overstate the conclusions.

I suggest the authors consider refocusing the manuscript to address the two most valuable contributions of the paper 1) the need to be more holistic in considering both work and non-work “loads” and recovery (this point is well made) and 2) use most of the paper to discuss lessons learned from the application of EMA in TWH research as a methodologic advance.   Being a pilot, with many limitations, I recommend that the authors de-emphasize any conclusions about relationships between workload/recovery x well-being.

In the introduction, without formally stating a hypothesis, the authors propose two aims for their study: 1) understand the relationship of whole-day activity 74 to worker well-being, and 2) provide preliminary evidence for ecological momentary assessment (EMA) as a valid method for investigating workers’ exposures to high workload and recovery activities across work and non-work periods. I see that when we get to lines 181-183, specific testable hypotheses are stated for the first time: “As shown in Table 2, we expected that, 1) frequency of high-workload activities  would be positively associated with end of day workload and negatively associated with well-being [15], and 2) frequency of recovery activities would be negatively associated with workload [9] and positively associated with well-being [41,42].”  It would help the reader to see these stated in the introduction.

It would be helpful for the authors to discuss and reference relevant theoretical framework(s) that provide the basis for their study. As part of this, they should also discuss possible mediators/moderators of the relationship between workload/recovery x well-being. There are many.

Lines 81-84 I have concerns about generalizability. The authors assert that their findings are generalizable because of the prevalence chronic health conditions in the workforce, including an 11% prevalence of diabetes. Generalizing on the experience of people who have a relatively rarer and usually lifelong form of diabetes (i.e. type 1) and inferring that these 45 individuals’ data can be generalized to all forms of diabetes, and to all other ‘chronic conditions’ is a stretch. I’d be interested in what the literature has to say about the similarities and differences in emotional and functional status of those with type 1 v. type 2 diabetes before asserting generalizability, even for just that part of the generalizability statement. Generalizability should be discussed in a limitations section of the Discussion.

The authors collected well-being information at the same time as information about workload and recovery, which presumably would lead individuals to readily be able to infer the purpose of the study. How do the authors propose to mitigate the likely biases introduced by this method of data collection? One of my most fundamental concerns with the design (and hence conclusions) of the paper has to do with what happens when people complete a series of questions pertaining to both the dependent and independent variables in a single survey (and in fact do so repeatedly). The authors took steps to help mitigate the risk of inflated intercorrelation coefficients etc (eg. using end-of-day questions), but there remains a strong likelihood that the intercorrelation coefficients may be inflated. Was there some attempt to alter the order of questions to mitigate order effects? Did they consider asking about outcomes (e.g. well-being) separately from questions that measured exposure (e.g. task load)?  What other strategies might help researchers who attempt similar studies to avoid pitfalls related to survey item order? 

They conducted EMA data collection 5-6 times per day over a 14-day period in a cohort of 45 workers with type 1 Diabetes.  The authors use five single survey items on their EMA to assess well-being: stress, fatigue, pain and positive affect/negative affect. These roughly align with some of the dimensions found in the NIOSH WellBQ as they point out, and with elements of some other well-being instruments as well. Well-being is increasingly understood to be multidimensional and to probably require more than a partial set of domains represented by single-item measures to get at the subdomains. For certain, there is a great need for more single-item measures in this area of research (see for example: Fisher, Matthews and Gibbons J Occ Health Psychology 2016). Especially when doing EMA multiple times a day, I appreciate the need to reduce respondent burden (and possibly even improve face validity). The single-item measures used to characterize well-being in the current study require further justification. Can the authors provide evidence from the literature that these single items have sufficient validity relative to other measures of well-being or even one or more of its domains?  Additionally, do the authors have concerns about the other dimensions of well-being (e.g. those other dimensions found in the WellBQ) that they did not assess? This should be addressed in the discussion and as a limitation.

Activity Exposure measures: The authors indicate that a measure called “total task load” is the simple sum of (I think) six task loads scored individually. Can the authors provide evidence from the literature or other rationale for a summing of these and for treating them as having equal weights?  I ask this because for some jobs/duties the weight of one or another task may be greater than for others. In fact, there is a line of reasoning that to understand task load, it is also necessary to understand load in a context connecting workers’ abilities (job difficulty) to perform cognitively and/or physically demanding jobs, as well as underlying chronic health conditions. (See for example Jinnett K, et al, in Health Affairs (Millwood) 2017.) 

In the process of “conceptually sorting” (dichotomizing) their two major exposure variables into “high workload” and “recovery,” the authors make a set of assumptions that may or may not be valid. They address these limitations in the methods section in lines 122-150. Much of this could be moved to the Discussion and addressed under an expanded limitations section, along with a more balanced examination of the pros and cons of their assumptions and approach.

Perceived Workload. The authors make an excellent effort to try to validate their workload estimates. In the limitations section of the discussion (along with the paragraph above this one) please clarify what is meant by an “adapted version” of the NASA-TLX. What parts of the NASA-TLX were used, exactly? As a general comment, elsewhere in the methods section of this paper  (e.g. section 2.2.3) the authors refer to the use of selected items and say that they ‘adapt” from other survey instruments. Please provide greater clarity of what this means, and (in the limitations section) how “adapting” might affect their ability to assert that they have truly validated their other instrument against an adapted validated instrument. It is possible but unlikely that they can assert that by adapting another instrument they retain its validity. I normally would not be this picky, but because the authors are heavily relying on this to defend their approach to constructing their major exposure measures, this should be addressed.

It is unclear how much of the scale development and the decisions about how to construct scores were done a priori vs. post hoc.  Please state if the development of the various exposure measures, and the calculation of individuals’ exposure and well-being scores, were done prior to reviewing the data set, and, also if the scoring for individuals’ various measures were done blind to the outcomes of interest, demographics, etc.  On a related point, when were the decisions made to perform the three mixed models? Was that decided at the time of study design, or post hoc? Were there additional mixed models examined in addition to the three included in this report? I’m trying to understand how much of the study was exploratory in nature.

L 191-199. On lines 195-196 the authors state: “Instead of adjusting for the multiple statistical tests with statistical corrections 195 of alpha, we report 95% confidence intervals for all statistical tests .” While I agree that conventional approaches to mitigating the effect of multiple comparisons when testing for statistical significance can be overly conservative, my enthusiasm for the analysis is dampened by a large number of within-person correlations (n = 21), the number of mixed models (n = 3) conducted, and the inclusion of both 95% CIs and p values. In Table 2, for example, the authors seem to want to have it both ways. To be internally consistent, I would discourage the authors from including p values and refrain from using terms like “significant associations”. (l. 244 for example), especially in the results section. They may want to defend their findings as “significant” in the discussion and limitations section. I would also recommend that a full intercorrelation matrix be included either in the body of the manuscript or as an appendix.

Discussion and Conclusions: For the reasons mentioned above, I would encourage the authors to take a different approach to the framing of the paper and the Discussion. I would suggest presenting it as a pilot (exploratory) study that examines the potential use of EMA to assess these rather complex constructs and explore the strengths and weaknesses of the approach. I would de-emphasize the “significance” of the findings.

Limitations paragraph:  I greatly appreciate the thoughtful comments provided in the limitations section. This section should be expanded. Or, if the authors take my suggestion and refocus the paper as a pilot application and discussion of the methodologic value of EMA in TWH research (and not on the purported relationship of exposures x well-being outcomes), the limitations section could be turned into a major part of the paper, examining lessons learned from the use of EMA.

Reviewer 3 Report

This research investigates worker well-being relative to high-workload and recovery activities. Specifically, the authors examined that how frequencies of high-workload and recovery activities from both work and non-work periods are associated within a day. Overall, the paper deals with a very interesting and important topic. The manuscript is well written and structured. I only have a few suggestions for further improvement of the paper.

It is quite predictable that well-being and workload are associated. Thus, the authors should effectively deal with an originality issue of the research. The authors should demonstrate what the originality and value of their research, and clearly state it in the manuscript. 

A theoretical implication is another concern. While the authors center on emphasizing the practical implications of the research, its theoretical meaning is weakly stated.

A professional editing is required. I have found several grammatical/editorial errors while reading the manuscript. In addition, some sentences are awkwardly written. 

There are many recent studies related to the topic of this research. I suggest the author(s) to include more recently published papers related to the topic. The followings are the sample articles dealing with the same topic. For the improvement of the manuscript quality, integrating the following studies into the manuscript are recommended:

Ariza-Montes, A., Hernández-Perlines, F., Han, H., Law, R. (2019). Human dimension of the hospitality industry: Working conditions and psychological well-being among European servers. Journal of Hospitality and Tourism Management, 41, 138-147.

Radic, A., Arjona-Fuentes, J.M., Ariza-Monthes, A., Han, H., Law, R.  (2020). Job demands–job resources (JD-R) model, work engagement, and wellbeing of cruise ship employees. International Journal of Hospitality Management, 88, 102518

Round 2

Reviewer 2 Report

The authors have done an outstanding job of responding to earlier critiques of the manuscript. The result is a paper that provides a sound, well-articulated case for the use of a novel technique for assessing workload. They make a strong case for us to broaden the scope of our measures to consider both work and non-work conditions. They provide a sound case for how their work fits (and advances the thinking) about existing theoretical models. The shift of emphasis plus added information about the methods are valuable contributions. The interpretation of the pilot data is appropriate, given the limitations of cohort size. I wish to compliment the authors for the high quality of this revised manuscript.

Two minor suggestions:

It is important to include the information provided on lines 115-134, but most of this information would be better suited for inclusion in the Discussion under Limitations, to make their case for generalizability.  A sentence or two in the Methods is fine to explain why the authors took this approach, but I'd suggest that they save the opinion for later.

I would suggest rewording the rather complex, grammatically challenging sentence on lines 468-470. Perhaps it would read better as two or three separate sentences.

Reviewer 3 Report

The authors have considerably improved the manuscript. The revised version looks much better than the original one.  

Author Response

Thank you again for your feedback in the first round. In terms of the moderate English changes required, we again will defer to the editor regarding the need for further editing of English language used.